# Multi-Omics Revealed Peanut Root Metabolism Regulated by Exogenous Calcium under Salt Stress

**DOI:** 10.3390/plants12173130

**Published:** 2023-08-31

**Authors:** Xuan Dong, Yan Gao, Xuefeng Bao, Rongjin Wang, Xinyu Ma, Hui Zhang, Yifei Liu, Lanshu Jin, Guolin Lin

**Affiliations:** 1College of Land and Environment, Shenyang Agricultural University, No. 120 Dongling Road, Shenhe District, Shenyang 110866, China; 2019200115@stu.syau.edu.cn (X.D.); gaoyan2915@stu.syau.edu.cn (Y.G.); baoxuefeng2xichang@163.com (X.B.); 2021220451@stu.syau.edu.cn (R.W.); huizhang@syau.edu.cn (H.Z.); liuyifei@syau.edu.cn (Y.L.); jinlanshu@syau.edu.cn (L.J.); 2Testing Center for Agricultural Product Safety and Environmental Quality, Shenyang Institute of Applied Ecology, Chinese Academy of Sciences, No. 72, Culture Road, Shenhe District, Shenyang 110017, China

**Keywords:** abiotic stress, calcium, metabolism, multi-omics, salt stress, peanut (*Arachis hypogaea* L.), phytohormone

## Abstract

High salinity severely inhibits plant seedling root development and metabolism. Although plant salt tolerance can be improved by exogenous calcium supplementation, the metabolism molecular mechanisms involved remain unclear. In this study, we integrated three types of omics data (transcriptome, metabolome, and phytohormone absolute quantification) to analyze the metabolic profiles of peanut seedling roots as regulated by exogenous calcium under salt stress. (1) exogenous calcium supplementation enhanced the allocation of carbohydrates to the TCA cycle and plant cell wall biosynthesis rather than the shikimate pathway influenced by up-regulating the gene expression of antioxidant enzymes under salt stress; (2) exogenous calcium induced further ABA accumulation under salt stress by up-regulating the gene expression of ABA biosynthesis key enzymes AAO2 and AAO3 while down-regulating ABA glycosylation enzyme UGT71C5 expression; (3) exogenous calcium supplementation under salt stress restored the *trans-*zeatin absolute content to unstressed levels while inhibiting the root *cis*-zeatin biosynthesis.

## 1. Introduction

Soil salinization is a critical environmental issue that poses a significant threat to agricultural production. High sodium ion concentrations in the environment result in potassium outflow from plant cells, resulting in an ion distribution imbalance between plant cells [1]. As well as causing secondary oxidative stress and ROS burst in plant cells [2], it also causes cytoplasmic membrane peroxidation and electrolyte leakage [3]. Salt stress can disrupt hormonal homeostasis in seedlings [4], leading to decreased levels of endogenous phytohormones such as indole-3-acetic acid [5] and cytokinin [6], which impair plant metabolism and development. Salt stress can also affect plant photosynthesis [7]. Around 20% of cultivated land and 50% of irrigated land are at risk of salinization worldwide [8,9]. A majority of high-quality crops are glycophytes [10], which lack effective mechanisms to tolerate high salt concentrations. Finding feasible ways to improve crop salt tolerance is essential for agricultural sustainability.

Calcium, an essential plant nutrient element and second messenger, has been shown by numerous studies to play an essential role in improving plant salt tolerance. Exogenous application of nano-calcium phosphate (CaP-NPs, nano-fertilizer) can significantly increase the yield of faba bean in saline soil [11]. Calcium-rich animal manure (Ca-FC) in combination with fertilizer effectively reduced adverse effects of salinized soil on the growth of oilseed rape as well as increased plant N, P, and K uptake, increasing the seed yield and thousand grain weight [12]. Exogenous calcium increased the rice seed germination rate in NaCl solutions [13], significantly improved the activities of SOD, CAT, POD, and GR in tartary buckwheat seedling leaves under salt stress, and increased the contents of AsA and GSH content, while decreasing MDA [14] under salt stress. It enhanced the chlorophyll A content and significantly reduced electrolyte leakage in red amaranth seedlings [15], significantly increasing the wheat seedling leaves net CO_2_ assimilation rate, transpiration, stomatal conductance, and water use efficiency under salt stress. SOS3 (CBL4) binds to SOS2 (CIPK24), inducing the activation of SOS2, which in turn activates SOS1 (Na+/H+ antiporter) to expel Na^+^ from cells [16]. The overexpression of genes encoding for calcium signaling repeater proteins or calcium transporter elements has also been reported to improve salt resistance in plants [17]. The overexpression of CPK1 [18] and CPK12 [19] improves salt resistance in Arabidopsis. The overexpression of VaCPK21 improves salt resistance in grape [20]. Rice salt stress resistance can be enhanced by OsACA6 overexpression [21]. Recent studies found that TaNCL2-A overexpression can enhance Arabidopsis’ salt tolerance [22]. The salt tolerance of apples can be increased by the overexpression of MdCCX2 [23]. The studies above have shown that calcium improves plants’ ability to tolerate salt.

Peanut (*Arachis hypogaea* L.) is a major oil crop in China. Cultivated peanuts are always salt-sensitive varieties [24], but exogenous calcium can effectively mitigate salt stress effects on peanuts. Exogenous Ca (NO_3_)_2_ significantly increased the main stem height, lateral branch length, branch number, root weight, and individual peanut plant biomass under salt stress [25]. Shi found that the application of 150 kg/hm^2^ CaO at a concentration of 0.3% salinized soil significantly improved the distribution ratios of nitrogen, phosphorus, calcium, and magnesium elements in peanut seeds subjected to salt stress, yield per plant and fruit weight at maturity [26], and the accumulation of dry matter in seeds and pods of 64.29% [27]. Exogenous calcium has been studied in peanuts for alleviating salt stress, but most of the studies have focused on leaf physiology, leaf photosynthesis, and the effects of calcium fertilizer combined with chemical fertilizer on peanut yield and agronomic traits in salinized soils. Root growth is critical for seedling development, and the root system is the tissue directly exposed to salt stress. However, the pattern of root metabolism during this process has not been reported.

Omic methods (transcriptome, proteome, metabolome, etc.) are more efficient than traditional experimental methods. The analysis of high-throughput testing data using bioinformatics methods can provide a more comprehensive understanding of biological issues [28]. The use of omics methods to study abiotic stress has grown rapidly. Anthocyanin gene expression and salt tolerance in maize were examined using transcriptomics [29]. It was studied whether exogenous melatonin alleviated drought stress in wheat by altering gene expression in the jasmonic acid biosynthesis pathway and the lignin biosynthesis pathway through transcriptomics [30]. A metabolism study was performed to analyze the change trend of glucose metabolism and the downstream TCA cycling substance content in rice under copper stress [31]. Each omics method has its own technical characteristics. Transcriptome focuses on gene expression changes, and metabolome focuses on metabolites changes. Multiple omics can reveal mechanisms of biological problems from different levels, and can also analyze a specific biological process in the test results to ensure that the results are reliable. We used physiological phenotypic traits to assess the effect of exogenous calcium on peanut seedling roots under salt stress. Then, we performed transcriptome sequencing, broadly targeted metabolome assay, phytohormone absolute quantification assay, and RT-qPCR assay to investigate metabolism in peanut roots regulated by exogenous calcium under salt stress.

Phytohormones metabolism regulates root growth. Comprehensive transcriptome and hormone quantification analyses are helpful for identifying phytohormones controlled by exogenous calcium under salt stress, as well as key genes involved in their biosynthesis and signaling. They can provide a basis for further understanding the complex regulation of exogenous calcium on plant root growth. Plant growth materials are provided by root primary and secondary metabolites. Classification studies on the biosynthesis of carbohydrates, amino acids, lipids, and secondary metabolites offer preliminary insights into the mechanisms by which exogenous calcium regulates plant root metabolism under salt stress. We hypothesized that: (a) the application of exogenous calcium under salt stress may increase the biosynthesis of hormones related to plant stress resistance; (b) exogenous calcium application under salt stress may improve the activity of antioxidant enzymes, leading to an increase in primary metabolites biosynthesis such as carbohydrates and amino acids, while inhibiting secondary metabolites biosynthesis in peanut seeding roots. The aims of this study are: (a) to determine key pathways regulated by exogenous calcium in peanut seedling roots under salt stress by transcriptome and broadly targeted metabolome combined analysis; (b) to examine the regulation pattern of carbohydrates, amino acids, C6-C3 compounds, and small molecular lipids in peanut roots under salt stress by analyzing the expression patterns of different substances under different treatments; (c) to study the regulation of exogenous calcium on the amount of hormone biosynthesis in peanut roots and the expression of key enzyme genes in the synthesis pathway under salt stress transcriptome and phytohormone absolute quantification combined analysis.

## 2. Results

### 2.1. Physiological Phenotypic Traits of Exogenous Calcium Alleviate Salt Stress in Peanut Seedling Roots

To intuitively show the exogenous calcium alleviation effect on peanut seedling roots under salt stress, firstly, we photographed peanut roots after 48 h of treatment (Figure 1a). Then, we measured root length (Figure 1b), root weight (Figure 1c), and root vitality (Figure 1d), which showed similar trends under different treatments. Root length, root weight, and root vitality were significantly inhibited under Na treatment compared to CK (*p* < 0.05), while, under Na_Ca treatment, they all significantly increased compared to Na treatment (*p* < 0.05).

Peanut seedling root Na^+^, K^+^, and Ca^2+^ contents under different treatments were measured. Na treatment significantly reduced root K^+^ content (*p* < 0.05) while significantly increasing Na^+^ content (*p* < 0.05) compared to CK treatment, while Na_Ca treatment significantly increased K^+^ content (*p* < 0.05) while significantly reducing Na^+^ content (*p* < 0.05) compared to Na treatment (Figure 1e,f). Na treatment significantly decreased root Ca^2+^ content (*p* < 0.05) compared to CK treatment, while Na_Ca increased root Ca^2+^ ion content but not significantly compared to Na treatment (*p* < 0.05) (Figure 1g).

To reflect secondary oxidative stress in roots, O_2_^−^, H_2_O_2_, and MDA contents under different treatments were measured. Correspondingly, three antioxidant enzyme activities were also measured. Na treatment significantly increased O_2_^−^ and MDA content in roots compared with CK treatment (*p* < 0.05), while Na_Ca treatment significantly decreased them (*p* < 0.05) compared with Na treatment (Figure 1h,j). There was no significant difference in root H_2_O_2_ under the four treatments (Figure 1i). Na treatment significantly decreased SOD and POD activity (*p* < 0.05) compared to CK, while Na_Ca treatment significantly improved CAT and POD activity (*p* < 0.05) (Figure 1l,m).

The above physiological phenotypic traits showed that exogenous calcium inhibited Na^+^ uptake and K^+^ loss; improved antioxidant enzyme activities; eliminated the ROS content; and reduced the MDA accumulation, alleviating salt-induced damage to the peanut root.

### 2.2. DEGs and DEMs Screening

Root samples after 48 h of each treatment were collected (12 samples in total) for RNA sequencing. In 12 sequencing libraries, raw reads were 45,015,158 at the minimum and 52,736,760 at the maximum. Clean read ratios ranged from 95.85 to 98.77%. Mapped reads varied between 86.82 and 97.25% (Appendix A). After read count quantification, principal component analysis (PCA) was utilized to project each sample point onto the first and second principal components for all sample count matrices (Appendix A). Differences between treatment groups can be identified and the biological repeatability of the samples in each treatment group can satisfy the requirements of the downstream data analysis.

To screen DEGs that are responsive to salt stress and can be regulated by exogenous calcium under salt stress, three pairwise comparison groups were established: Na vs. CK, Na_Ca vs. Na, and Ca vs. CK (treatment group vs. control group). EdgeR was used for differential expression analysis. DEGs were screened according to the three following criteria: for each pairwise comparison group, an average CPM value > 4 at least in one treatment; |log_2_FC| > 2; FDR < 0.05.

Na treatment resulted in 7962 genes differentially expressed (1679 up-regulated and 6283 down-regulated) compared with CK (Figure 2a). Na_Ca treatment resulted in 6209 genes differentially expressed (3049 up-regulated and 3160 down-regulated) compared with CK. Compared with CK, Ca treatment regulated 650 genes differentially expressed (287 up-regulated and 363 down-regulated). DEGs shared by three pairwise comparison groups were calculated (Figure 2b). A total of 3846 DEGs shared by two pairwise comparison groups (Na_Ca vs. Na and Na vs. CK), which should be considered DEGs involved in exogenous calcium mitigating salt stress, were analyzed for GO slim enrichment (Appendix A).

The broadly targeted metabolome assay was performed. A total of 1038 metabolites were detected in all samples. PCA was performed on the relative contents of metabolites in each sample and a mixture of samples (15 in total) and all sample points were projected onto a 2D scatter diagram (Appendix A).

We screened the DEMs according to the following steps. Firstly, three pairwise comparison groups were selected (Na vs. CK, Na_Ca vs. Na, Ca vs. CK) and then metabolites whose average value was greater than 1E+5 for at least one treatment were screened for each pairwise comparison group. Then, OPLS-DA analysis was performed to screen DEMs according to the following two criteria: |log_2_FC| > 1.5; *p* < 0.05.

Frequency statistics of the screened DEMs from the three pairwise comparison groups were performed. Compared with CK, Na treatment affected the expression of 196 metabolites differentially (75 up-regulated and 121 down-regulated) (Figure 2c). Na_Ca treatment affected the expression of 141 metabolites differentially (96 up-regulated, 45 down-regulated) compared with Na treatment. Ca treatment regulated the expression of 31 metabolites (14 up-regulated, 17 down-regulated) compared with CK.

The DEMS Venn diagram analysis (Figure 2d) showed that 108 DEMs were shared by two pairwise comparison groups (Na_Ca vs. Na and Na vs. CK), which represent DEMs salt stress alleviation by exogenous calcium. These DEMs were counted according to the classification of substances (class I, class II, and compound) as shown in Figure 2d, and their relative biosynthesis under each treatment can be seen in Appendix A.

Based on differential expression screening and Venn diagram analysis, we screened 3648 differentially expressed genes (DEGs) and 108 differentially expressed metabolites (DEMs). They were influenced by salt stress and can be regulated by exogenous calcium under salt stress.

### 2.3. KEGG Co-Enrichment Analysis of Differentially Expressed Genes and Differentially Expressed Metabolites

KEGG enrichment analysis was initially performed on DEGs and DEMs shared by two pairwise comparison groups (Na vs. CK and Na_Ca vs. Na), respectively. Then, common pathways in the DEGs and DEMs KEGG enrichment analysis results that enriched at least two DEMs were selected to count the DEGs and DEMs numbers for each pathway (Figure 3). Pathways were classified according to the three-level KEGG pathway classification (KEGG A, KEGG B, Pathway).

For the KEGG A class level, 22 pathways belonged to the metabolism class, 2 to the environmental information processing class, and 1 to the genetic information processing class (Figure 3). Therefore, pathways belonging to metabolism (KEGG A class) enriched the most DEGs and DEMs. For its subordinate class (KEGG B class), the carbohydrate metabolism class contained seven pathways, the amino acid metabolism class contained eight pathways, and the biosynthesis of other secondary metabolites class contained three pathways.

Notably, three pathways (phenylpropanoid biosynthesis, flavonoid biosynthesis, and isoflavonoids biosynthesis) belonging to the biosynthesis of the other secondary metabolites class (KEGG B Class) are downstream metabolic pathways of the phenylalanine metabolism pathway (belonging to the KEGG B class amino acid metabolism). Moreover, among the 25 pathways, the plant hormone signal transduction pathway enriched the most DEGs.

The results of the KEGG co-enrichment analysis of DEGs and DEMs showed that exogenous calcium supplementation under salt stress can regulate the expression of enzyme genes and the biosynthesis of metabolites involved in carbohydrate, amino acid, and secondary metabolism. Meanwhile, a large number of DEGs were regulated in the plant hormone signal transduction pathway, which is related to environment information processing.

### 2.4. Comprehensive Analysis of DEMs and Antioxidant Enzyme DEGs

Plant metabolites are diverse, and the KEGG annotation cannot cover all DEMs detected by the broadly targeted metabolome. In light of this, it is necessary to present changing trends in carbohydrates, lipids, C6-C3 plant secondary metabolites, amino acids, and their derivatives according to their substance classification to reflect regulation by exogenous calcium under salt stress. Considering that C6-C3 plant secondary metabolites (phenols and flavonoids) are non-enzymatic antioxidants, the comparison of their biosynthesis to the gene expression of antioxidant enzymes can reflect, to some extent, the plants antioxidant change trend under different treatments.

The mean relative biosynthesis of saccharides and alcohols, liquids, amino acids, and derivatives under different treatments is shown in Figure 4. Among them, the vast majority of amino acids and derivative DEMs and saccharide and alcohol DEMs were expressed in the same pattern under different treatments: Na treatment down-regulated their biosynthesis compared to CK treatment, while Na_Ca treatment up-regulated their biosynthesis compared to Na treatment. The vast majority of the detected liquid DEMs (including free fatty acids, LPCs, and LPEs subclass) showed a consistent expression pattern under treatments: Na treatment up-regulated their biosynthesis compared to CK treatment, while Na_Ca treatment down-regulated their biosynthesis compared to Na treatment. There were also a few DEMs with opposite expression patterns to other DEMs in this class, including 3-methyl-1-pentanol, thymine, ribosyl-adenosine, and 7S,8S-DiHODE. The relative biosynthesis of these DEMs in each sample is shown in Appendix A.

The above results showed that salt stress up-regulated three types of lipid (LPE, LPC, and FFAs) biosynthesis and down-regulated saccharides and alcohols, amino acids, and their derivatives biosynthesis, while exogenous calcium down-regulated these three types of lipids biosynthesis and up-regulated saccharides and alcohols, amino acids, and their derivatives’ biosynthesis under salt stress.

The expression of DEGs encoding peroxidases, catalases, and superoxide dismutases shared by two comparison groups (Na vs. CK and Na_Ca vs. Na) as well as the biosynthesis of phenolic acids and flavonoids DEMs under different treatments is shown in Figure 5.

All of the catalase and superoxide dismutase superfamily and most of the peroxidase superfamily DEGs showed a consistent expression pattern under different treatments: Na treatment down-regulated the expression of these antioxidant enzyme DEGs compared to CK treatment, while Na_Ca treatment up-regulated their expression compared to Na treatment (Figure 5).

Meanwhile, 8 out of the 67 peroxidase superfamily DEGs showed the opposite diametrically expression pattern to other members under different treatments. These included: three PER5 (LOC112702699, LOC112737096, and LOC112789186), two PER47 (LOC112751597 and LOC112801773), one PER68 (LOC112766174), and two PNC1 (LOC112743807 and LOC112793786).

For phenolic acids and flavonoids, which are important non-enzymatic antioxidants, most DEMs exhibited diametrically opposed expression patterns to the antioxidant enzyme DEGs under different treatments. Also, we noted that 5 of the 20 differentially expressed phenolic acids showed the opposite expression pattern to others under different treatments. They were: 4-aminobenzoic acid, *cis*-coumaric acid, 2-acetyl-3-hydroxyphenyl-1-O-glucoside, 4-O-glucosyl-sinapate, and anthranilate-1-O-sophoroside.

The above results indicate that salt stress down-regulated the expression of genes encoding antioxidant enzymes in general while up-regulating the biosynthesis of phenolic acids and flavonoids, whereas exogenous calcium up-regulated the expression of antioxidant enzyme genes while down-regulating phenolic acids and flavonoids biosynthesis under salt stress.

### 2.5. Comprehensive Analysis of Differentially Expressed Genes and Metabolites in Phytohormone Biosynthesis and Signal Transduction Pathways

GO enrichment analysis (Appendix A) and KEGG co-enrichment analysis (Figure 3) suggest that phytohormones play a non-negligible role in the alleviation of salt stress by exogenous calcium. However, the broadly targeted metabolome assay cannot absolutely quantify phytohormone biosynthesis. We used the LC-MS/MS technique to quantify the absolute biosynthesis of 7 kinds of phytohormones (15 in total) in peanut roots after 48 h treatment. Meanwhile, a comprehensive analysis was performed in combination with transcriptome data (Figure 6).

The results of one-way ANOVA for 15 phytohormones absolute biosynthesis in peanut roots after 48 h treatment is shown in Figure 6c. Compared with other phytohormones, ABAs, Jas, and zeatins exhibited different patterns under different treatments. To some extent, they can be considered as key phytohormones involved in salt stress alleviation by exogenous calcium in peanut seedling roots.

ABA biosynthesis under Na treatment was significantly higher than CK (*p* < 0.05), while ABA biosynthesis under Na_Ca treatment was 3.34 times that of Na treatment, with a significant difference between the two treatments (*p* < 0.05) (Figure 6c). The ABA-GE biosynthesis in the two treatments without salt stress was significantly lower than in the salt stress treatment (*p* < 0.05), while there was no significant difference in ABA-GE biosynthesis between the Na and Na_Ca treatments (*p* < 0.05). Fourteen of the seventeen DEGs on ABA biosynthesis and signal transduction pathways had consistent expression patterns: compared with CK treatment, Na treatment down-regulated their expression, and, compared with Na treatment, Na_Ca treatment up-regulated their expression (Figure 6a). Three DEGs showed the opposite expression patterns in the two pairwise comparison groups, including two ABA2 (LOC112728114 and LOC112751370) and one PYR/PYL (LOC112702333).

Put another way, for ABA biosynthesis, exogenous calcium up-regulated AAO3 expression, ABA biosynthesis was further stimulated, and its glycosylation product ABA-GE was down-regulated under salt stress. Moreover, exogenous calcium up-regulated most DEGs involved in ABA signal transduction under salt stress.

Seven jasmonic acids and their derivatives (12-OPDA, JA, MeJA, JA-Val, JA-Phe, JA-Ile, H2JA) were detected (Figure 6c). Among them, the JA, H2JA, JA-Ile, and JA-Val biosynthesis trends under different treatments were noteworthy. JA biosynthesis under Na treatment was significantly greater than the other three treatments (*p* < 0.05), while JA biosynthesis under these three treatments did not significantly differ (*p* < 0.05). The H2JA biosynthesis change trend under different treatments was the opposite of JA. For JA-Ile, its biosynthesis under Ca treatment was significantly greater than the other three treatments (*p* < 0.05), and, under Na_Ca treatment, its biosynthesis was significantly higher than under Na treatment. For JA-Val, its biosynthesis under Na treatment was significantly greater than under the other three treatments (*p* < 0.05).

The unified expression pattern shared by most DEGs in JA biosynthesis (Na vs. CK and Na_Ca vs. Na) contrasted with most JA signal transduction DEGs (Figure 6a). For the 17 DEGs encoding the three key enzymes of JA biosynthesis, ACX, MFP2/AIM1, and ACA2, the expression patterns of 12 DEGs were consistent. Compared with CK treatment, Na treatment up-regulated their expression, and, compared with Na treatment, Na_Ca treatment down-regulated their expression. A total of 21 of the 23 JA signaling DEGs shared a unified expression pattern, which contrasted with the above pattern, except for 2 MYC2-encoded DEGs (LOC112732736 and LOC112790889). Additionally, for five DEGs involved in JA biosynthesis, four DEGs encoding MFP2/AIM1 (LOC112741727, LOC112796415, LOC112751218, and LOC112802129), and one DEG encoding ACA2 (LOC112727727), Na treatment down-regulated their expression compared to CK treatment, while Na_Ca treatment up-regulated their expression compared to Na treatment.

Put another way, for the JA biosynthesis pathway, exogenous calcium under salt stress up-regulated JA biosynthesis substrate 12-OPDA, down-regulated JA biosynthesis, and down-regulated the expression of most DEGs encoding key enzymes of JA biosynthesis. In the JA signal transduction pathway, the biosynthesis of JA-Ile was up-regulated, while the expression of most DEGs was up-regulated.

Two kinds of zeatin, *cis*-zeatin (*c*Z) and *trans*-zeatin (*t*Z), were detected (Figure 6c). For *t*Z, biosynthesis under Na treatment was significantly higher than under the other three treatments (*p* < 0.05). The *c*Z biosynthesis change trend under different treatments was just the opposite to *t*Z. There were no significant differences in *t*Z and *c*Z contents between Na_Ca and CK treatments (*p* < 0.05).

A total of seven DEGs were enriched in the zeatin biosynthesis and signal transduction pathways. Among them, the expression patterns of five DEGs were consistent: Na treatment down-regulated their expression compared with CK treatment, while Na_Ca treatment up-regulated their expression compared with Na treatment (Figure 6a). For zeatin biosynthesis, Na treatment down-regulated two IPT DEGs (LOC112705443 and LOC112722977) compared with CK, and Na_Ca treatment up-regulated these two DEGs compared with Na treatment. For the zeatin signal transduction pathway, except for two ARR-A DEGs, the expression patterns of the other three were down-regulated by Na treatment in comparison with CK, and up-regulated by Na_Ca in comparison with Na treatment.

Put another way, compared with conditions without salt stress, exogenous calcium restored *t*Z and *c*Z biosynthesis under salt stress. Exogenous calcium up-regulated gene expression in zeatin biosynthesis and signal transduction pathways under salt stress.

### 2.6. RT-qPCR Verification

To verify the RNA-seq accuracy, we selected 14 DEGs for the RT-qPCR assay. Then, we compared whether their expression trends in two pairwise comparison groups (Na vs. CK and Na_Ca vs. Na) were consistent using these two quantification methods (Appendix A). These genes are involved in the biosynthetic and transformation pathways of ABA and JA, respectively.

Six genes encode key enzymes in the JA biosynthesis pathway. They are LOC112802428 (ACX1), LOC112720783 (ACX1), LOC112733299 (ACX2), LOC112791264 (ACX2), LOC112703784 (MFP2), and LOC112723093 (MFP2). Their expression trends in the two pairwise comparison groups were consistent between the two quantitative methods. Three genes (LOC112733726, LOC112758416, and LOC112695332) encode JAR1, the enzyme that couples JA to isoleucine to form a highly active signaling molecule JA-Ile. Regarding two JAR1-encoding genes (LOC112695332 and LOC112733726), the expression trends in the two pairwise comparison groups were inconsistent between the two quantitative methods.

There are three genes’ key enzymes in the ABA biosynthesis pathway. They are LOC112765030 (AAO3), LOC112702199 (AAO2), and LOC112765029 (AAO2). Their expression trends in the two pairwise comparison groups were consistent between the two quantitative methods. Two genes encode UGT71C5 (LOC112702879 and LOC112764382), whose function is to glycosylate ABA. The expression trends of the two genes in the two pairwise comparison groups were consistent between the two quantitative methods.

Twelve of fourteen genes expression trends in the two pairwise comparison groups were consistent between the two quantitative methods. Combining Appendix A and Figure 6c, exogenous calcium induced further ABA accumulation under salt stress by up-regulating the expression of ABA biosynthesis key enzyme genes AAO2 and AAO3 while down-regulating ABA glycosylation enzyme gene UGT71C5. Additionally, exogenous calcium down-regulated the gene expression of ACX1, ACX2, and MFP2, the key enzymes of JA biosynthesis, leading to decreased JA biosynthesis under salt stress.

## 3. Discussion

Excessive Na^+^ uptake leads to membrane depolarization and the activation of K^+^ efflux antiporters, resulting in K^+^ efflux [32]. Meanwhile, Na^+^ competes with K^+^ for binding sites on intracellular enzymes, leading to enzyme inactivation and the inhibition of plant metabolism within the cell [33]. Therefore, reducing Na+ uptake and K+ efflux under salt stress is crucial for maintaining normal plant metabolism. In the physiological phenotypic traits analysis, we found that exogenous calcium inhibited Na^+^ absorption and K^+^ efflux in the roots of peanut seedlings. These results were consistent with studies on rice [34], sour jujube [35], and Limonium stocksii [33]. By mining transcriptome data, we found DEGs related to Na^+^ and K^+^ transport (Appendix A). Exogenous calcium down-regulated two nonselective ion channels of CNGC-encoding DEGs (LOC112697431 and LOC112771534) under salt stress, which might contribute to the decrease in Na^+^ uptake under salt stress. Meanwhile, all DEGs encoding the specialized potassium transporter (POT) and one two-pore potassium channel (TPK1) DEG were also restored to unstressed levels. This might contribute to an increased K^+^ uptake and maintained intracellular K^+^ concentration under salt stress. The results above indicate that exogenous calcium maintains Na^+^/K^+^ homeostasis under salt stress by regulating the expression of genes coding for Na^+^ and K^+^ transport-related proteins.

It is well known that the antioxidant enzyme system can scavenge reactive oxygen species, which plays a key role in plant resistance to stress. Previous studies have shown that the overexpression of CuZnSOD [36], TaGR [37], and GhCAT1 [38] enhanced salt tolerance in tobacco, bread wheat, and cotton, respectively. In this study, exogenous calcium stimulated the activity of SOD, POD, and CAT enzymes, as well as up-regulated the expression of their coding genes (including: 59 PERs, 3 GPX6, 3 APXs, 3 CATs, 6 CDSs, and 2 FDSs). Further, by mining DEGs in transcriptome data, we also found that exogenous calcium under salt stress up-regulated the expression of five CMLs and four CDPKs (Appendix A). Previous studies have shown that the overexpression of CDPKs and CMLs-encoding genes can up-regulate plant antioxidant enzyme genes, which improves abiotic stress tolerance in plants. Citrus aurantium transgenic with the PtrCDPK10 gene overexpresses the APX gene and reduces ROS accumulation under drought stress [39]. The overexpression of the MdCPK1a gene increases the expression of genes encoding SOD and CAT, as well as the resistance of plants to low temperatures, droughts, and salt stresses [40]. ShCML44 overexpression enhances antioxidant enzyme activity under drought stress [41]. In Arabidopsis, the catalase CAT3 can be phosphorylated by CPK8 and enhance its activity, thus regulating ROS homeostasis [42]. Based on the above reports and analyses, we speculate that exogenous calcium could up-regulate antioxidant enzyme gene expression by up-regulating CPKs and CMLs under salt stress.

LPEs, LPC, and FFA biosynthesis is affected by antioxidant enzyme activities. We found that exogenous calcium down-regulated their biosynthesis under salt stress. It was reported that LPEs and LPC can be hydrolyzed by phospholipase D (PLD) into phosphatidic acids (PAs), or converted to free fatty acids (FFAs) by phospholipase hydrolysis A (PLA). Some studies reported that LPEs, LPCs, and FFAs accumulate under salt stress [43,44,45]. The plasma membrane phospholipids are hydrolyzed into LPEs and LPCs when secondary oxidation stress is caused by salt stress. Therefore, the up-regulation of the gene expression (and activity) of various antioxidant enzymes by exogenous calcium under salt stress, resulting in the alleviation of oxidative stress under salt stress, is the main reason for their reduced biosynthesis.

The carbon allocation of carbohydrate metabolism in plants under salt stress is also influenced by antioxidant enzyme activities. Glycolysis connects plant primary metabolism (TCA cycle) and secondary metabolism (shikimate pathway). The shikimate pathway is the upstream pathway of phenylpropane biosynthesis and flavonoid biosynthesis, and the carbon skeleton sources of 3-dehydlroshikimate are phosphoenolpyruvate (PEP) or β-D-fructose 6-phosphate in the glycolysis pathway. If glycolysis metabolic products are used to synthesize phenolic acid and flavonoids through the shikimate pathway, substrates allocated to the TCA cycle are inevitably reduced, which will result in the plant not maintaining its normal primary metabolism. Antioxidant properties of phenolic acids and flavonoids are derived from hydroxyl groups in their structures. Plants will only attempt to eliminate ROS through secondary metabolites biosynthesis, such as phenolic acids and flavonoids, when antioxidant enzyme activities have been exhausted. This is an inefficient process that crowds out carbon sources for primary metabolism. The improvement in antioxidant enzyme activities by exogenous calcium under salt stress reduces phenolic acid and flavonoid biosynthesis, whereas more glycolysis metabolic products can be allocated to the TCA cycle than to the shikimate pathway.

Relevantly, some DEMs in Appendix A are noticeable. We also noticed that some metabolites associated with the TCA cycle (such as oxalic acid, pimelic acid, monomethyl succinate, citric acid, and isocitric acid) were up-regulated under salt stress by exogenous calcium. Aspartic acid and its downstream derivatives (such as: L-aspartic acid, DL-methionine, L-methionine, glycyl-tryptophan, and S-(methyl)glutathione) were up-regulated under salt stress by exogenous calcium. The substrates for aspartic acid biosynthesis can be glutamine or aceto-oxalate in the TCA cycle, as well as acetyl-CoA in the glycolysis pathway. Aspartic acid is the substrate of many amino acids, such as methionine, methionine, cysteine, isoleucine, methionine, glutamate, and glycine [46], which are important for maintaining plant growth. Studies on exogenous SA [47] alleviating salt stress also obtained similar results. This can also be regarded as supplementary evidence that exogenous calcium affects the allocation of carbohydrates in the glycolysis pathway of peanut roots under salt stress.

Related to this is the fact that the biosynthesis of five carbon sugars and six carbon sugars (such as D-arabinose, L-xylose, D-mannose, D-glucose, and D-galactose) was up-regulated by exogenous calcium under salt (Figure 4), and the cell-wall-related GO term was also significantly enriched in transcriptome GO enrichment analysis (Appendix A). These five carbon sugars and six carbon sugars can be used as substrates for hemicellulose biosynthesis. Hemicellulose is one of the main components of the cell wall [48], which protects cells against ion toxicity [49]. In summary, exogenous calcium influences carbon allocation in plant carbohydrate metabolism by improving antioxidant enzyme activities under salt stress, which causes more carbohydrates to be used as substrates for the TCA cycle and cell wall biosynthesis than for secondary metabolism.

Phytohormones are significant active substances in plants. Usually, a variety of hormones in plants collaborate to regulate plant roots development and the response to various stresses. The biosynthesis patterns of zeatin, jasmonic acid (JA), and abscisic acid (ABA) in peanut seedling roots regulated by exogenous calcium under salt stress were different. For instance, exogenous calcium up-regulated IPT genes (LOC112705443 and LOC112722977) expression under salt stress. Exogenous calcium under salt stress restored the absolute content of *t*Z to a stress-free level while inhibiting the biosynthesis of its isomer *c*Z, which was also restored to the content level in the absence of stress. It was reported that cytokinin signaling promotes root growth; however, cytokinin signaling negatively regulates plant salt tolerance [50]. The mutants overexpressing IPT8, a key enzyme in cytokinin synthesis, were unable to scavenge ROS effectively under salt stress since antioxidant enzyme gene expressions were down-regulated [51]. Exogenous calcium increased antioxidant enzyme activities under salt stress, resulting in the normalization of IPT8 expression. We also noticed that exogenous calcium could further increase IPT8 expression compared with unstressed treatment (Appendix A). This indicates that IPT8 is positively regulated by Ca^2+^ and negatively regulated by Na^+^. As an isomer of *t*Z, *c*Z biosynthesis under different treatments was opposite that of *t*Z, which is a highly bioactive cytokinin [52]. When antioxidant enzyme activities are not sufficient to counteract salt-stress-induced secondary oxidative stress, plants will have to biosynthesize more non-enzymatic antioxidant substances and more *c*Z instead of *t*Z. In a related study on maize, salt stress increased *c*Z while decreasing *t*Z [53], which is similar to our results. The balance between *c*Z and *t*Z biosynthesis may provide a crucial signal for plants to decide whether to grow in salt-stressed environments.

Exogenous calcium under salt stress down-regulated the expression of most JA bio-synthesis key enzyme genes and JA absolute content, while up-regulating JAIle synthesis.

Under salt stress, exogenous calcium has differential effects on the expression of DEGs encoding JAR1 and MYC2, with some genes being up-regulated and others being down-regulated. We also noticed that exogenous calcium up-regulated the total count number of four DEGs encoding JAR1 under salt stress (Appendix A), which should be responsible for the up-regulation of the absolute JA-Ile content by exogenous calcium under salt stress. Similarly, we also noted that exogenous calcium up-regulated the total count number of six DEGs encoding MYC2 under salt stress, which implies that MYC2 should be down-regulated under salt stress. MYC2 is the central molecule of the JA signaling pathway. It has multiple functions in regulating plant root growth flexibly. On one hand, MYC2 binds to the G-box region of the PTL1/2 promoter region and negatively regulates the transcription of PTL1/2 (AP2/ERF encoding auxin response factor), thereby inhibiting plant main roots growth [54]. Alternatively, the overexpression of MYC2 also promotes lateral root growth [55]; that is, exogenous calcium relieves peanut root growth limitations caused by JA signal transduction under salt stress.

We found that exogenous calcium supplementation further results in ABA accumulation in peanut roots under salt stress, while ABA-GE biosynthesis is decreased. The results of RT-qPCR assays showed that this should be attributed to the fact that exogenous calcium can up-regulate the key enzyme gene for ABA biosynthesis, AAO2/3, while down-regulating the key enzyme gene for ABA glycosylation, UGT71C5, under salt stress. Analogously, some previous studies have also shown that exogenous calcium can lead to further ABA accumulation under salt stress [56,57,58,59]. As a result of salt stress, ABA biosynthesis is increased in plants [60]. Furthermore, ABA plays a key role in improving plant salt tolerance (or salt stress adaptation). ABA supplementation improves salt tolerance in Chlamydomonas reinhardtii [61], indica rice [62], and Tartary buckwheat [63]. According to some comparative studies of plant salt tolerance using salt-tolerant and salt-sensitive varieties as test materials, ABA levels in salt-tolerant varieties were significantly higher than in salt-sensitive varieties under salt stress [64,65,66]. It has been reported that some calcium signal receptors can up-regulate ABA biosynthesis in response to salt stress. Under salt stress, TaCDPK9 in wheat can affect TaNCED2 transcription by activating the transcription factors bZIP8, 9, and 13, resulting in ABA accumulation in wheat roots [67]. Overexpression of GHANN1, a calcium-dependent membrane-binding protein, enhances NCED expression in leaves under salt stress, improving cotton tolerance to salinity [68]. Overexpression of OsCaM1-1 up-regulated NCED and AAO gene expression in leaves, resulting in ABA accumulation under salt stress, enhancing rice salt tolerance [69]; that is, calcium can positively regulate ABA biosynthesis under salt stress through various mechanisms, and the exogenous calcium-mediated improvement of salt tolerance in plants should be partly related to ABA accumulation.

## 4. Materials and Methods

### 4.1. Experimental Materials Cultivation and Experimental Design

Haihua No. 1, a peanut cultivar widely distributed in China, was selected and purchased at Taobao Online Mall. Peanut seeds with uniform size and plumpness were rinsed thrice in distilled water, and then soaked in distilled water at 38 °C for eight hours. After swelling, peanut seeds were placed evenly in trays, covered with wet towels, and cultured at 29 °C in the dark for three days, with clean towels replaced daily. Then, selected individuals with roots more than twice the length of the long axis of the seed were randomly placed in plastic containers and cultured at 25 °C in the dark for 4 days. Individuals with uniform root growth and fibrous roots were selected for experimental treatment.

There were 4 treatment groups in this study: distilled water (CK); 150 mmol/L NaCl (Na); 15 mmol/L CaCl_2_ (Ca); and 150 mmol/L NaCl + 15 mmol/L CaCl_2_ (Na_Ca). Each treatment was biologically replicated 3 times, with 10 peanut seedlings per biological repetition.

For each treatment, peanut seedling roots were soaked in the solution, with stems above the liquid level. After 48 h treatment, plant roots were collected for physiological phenotypic traits assay, RNA-seq sequencing, broadly targeted metabolome assay, phytohormone absolute quantification assay, and RT-qPCR assay.

### 4.2. Experimental Materials Cultivation and Experimental Design

#### 4.2.1. Root Vitality

Plant root vitality was determined by 2,3,5-triphenyltetrazole colorimetry (TTC) [70], After measuring absorbance at 485 nm, root activity was converted according to the formula: reduction strength of tetrazole [mg/g (fresh weight of root)/h] = reduction amount of tetrazole (mg)/[root weight (g) × time (h)].

#### 4.2.2. Na^+^, K^+^ and Ca^2+^ Contents

A total of 0.1 g of dried plant material was accurately weighed, which was sieved through a 0.5 mm mesh, placed in a quartz crucible, and heated and carbonized at 300 °C for 30 min. Subsequently, the temperature was increased, and the sample was ashed at 500 °C for 2 h. The hot samples were dissolved in 1:1 (*v*/*v*) nitric acid, and up to 50 mL distilled water was added. Filtered liquids were collected in centrifuge tubes after filtering.

Flame photometry was used to determine sodium and potassium ion content [71]. A total of 10 mL of extraction solution was taken, 0.2 mL of 0.1 mol/L Al_2_(SO_4_)_3_ solution was added (to reduce Ca^2+^ interference with Na^+^ measurement), and then 10 mL (or 15 mL) of distilled water was added. Na^+^ and K^+^ content in samples were converted using NaCl and KCl standard solutions, respectively (mg/g D.M.). EDTA titration was used to measure calcium ion content [72]. For each sample, 5 mL was taken and the pH was adjusted to 7 with a NaOH solution. Then, 2 ml of triethanolamine solution (1:3, *v*/*v*) was added as a masking agent (chelating metal ions such as Al^3+^ and Mg^2+^ in the sample), and K-B indicator (potassium chrome blue: phenol naphthalene green B = m:m = 1:2.5) was used and titrated with 0.01 mol/L EDTA standard solution (EDTA-_2_Na). The reaction was considered complete when the solution changed from pink to blue. The Ca^2+^ in the plant samples was converted based on the reaction ratio between Ca and EDTA (mg/g D.M.).

#### 4.2.3. Superoxide Anion (O_2_^−^), Hydrogen Peroxide (H_2_O_2_) and Malondialdehyde (MDA) Content

Malondialdehyde content was determined by thiobarbituric acid (TBA) colorimetry under acidic conditions and converted to MDA content based on the fresh root mass. We diluted the freshly ground plant samples to 3 mL with 0.05 mmol/L phosphate buffer (pH 7.8) and added 5 mL of 0.5% thiobarbituric acid. After boiling for 10 min, the cooled sample tube was centrifuged for 15 min (3000 rpm/min) and the supernatant was collected. Absorbance values of the samples were determined at 532, 600, and 450 nm (0.5% thiobarbituric acid was used as a blank control).

To quantify superoxide anion, the hydroxylamine hydrochloride oxidation method [46] was used, and the absorbance at 530 nm was measured with an ultraviolet spectrophotometer, after which the sample O_2_^−^ content was converted using a standard curve. For the determination of hydrogen peroxide content, the peroxide–titanium complex precipitation method was adopted [73]. After the yellow precipitate was dissolved in 1 mmol/L H_2_SO_4_, absorbance at 410 nm was measured by an ultraviolet spectrophotometer. Then, the H_2_O_2_ content in the fresh sample was converted using the standard curve.

#### 4.2.4. Antioxidant Enzyme (SOD, CAT, POD) Activities

Plant roots enzymes were extracted with 50 mmol/L phosphate buffer solution (pH 7.8) and centrifuged at 12,000 rpm/min at 4 °C, and the supernatant was used to measure superoxide dismutase (SOD), catalase (CAT), and peroxidase (POD).

Superoxide dismutase (SOD) activity: after uniform exposure to light for 30 min using the nitroblue tetrazole (NBT) photoreduction method according to Giannopolitis and Ryes’ method [74], the sample absorbance was quickly measured by an ultraviolet spectrophotometer (adjusted to zero by the absorbance value of the liquid after standing in the absence of light for 30 min). An SOD activity unit was defined as the amount of enzyme required to inhibit 50% of NBT reduction in the fresh sample.

Cakmak and Marschner’s method was used to determine catalase (CAT) and peroxidase (POD) activity [75]. CAT activity was calculated by converting the CAT consumption rate per unit time after determining the absorbance value (reduction of H_2_O_2_) of the sample at 240 nm every 15 s. POD activity was extrapolated by measuring the rate of change in absorbance within 1 min (every 15 s) at 470 nm of the product formed by its reaction with guaiacol (o-methylphenol).

### 4.3. RNA Extraction, Library Construction, and RNA Sequencing

To extract total RNA from roots frozen at −80 °C, TRIzol reagent (Invitrogen, CA, USA) was used. Total RNA quantity and purity were determined using a Bioanalyzer 2100 and RNA Nano LabChip kit 6000 (Agilent, CA, USA). To purify polyadenylation mRNA from high-quality total RNA (Invitrogen), an oligo-T oligo-attached magnetic bead system (Invitrogen) was used. To reverse transcript the cleaved RNA fragments to create the final cDNA library, an RNASeq sample preparation kit (Illumina, San Diego, CA, USA) was used. The cDNA library was sequenced with a double-ended (150 bp) Illumina HiSeq 4000 system from LC Sciences (USA), in accordance with the protocol recommended by the supplier. Adapter readings and low-quality readings were removed from the original data for analysis. Clean reads were aligned with the reference genome (http://www.ncbi.nlm.nih.gov/data-hub/taxonomy/3818/ (accessed on 29 April 2021)) using HISAT2 software 2.2.1 [76] and transcripts were assembled according to alignment results. The original sequencing data were uploaded to NCBI (https://www.ncbi.nlm.nih.gov/bioproject/PRJNA964590 (accessed on 1 May 2023)).

### 4.4. Broadly Targeted Metabolome Assay

Root samples were freeze-dried and ground for 1.5 min at 30 Hz in a Tetsch MM 400 mixed grinder. To extract metabolites from the sample, 100 mg of grinding material was dissolved in 100 mL of 70% methanol aqueous solution, incubated overnight at 4 °C, centrifuged at 10,000× *g* for 10 min, and filtered using an SCAA-104 filter (0.22 μm pore size). A UPLC/MS/MS platform was used for metabolic analysis [77]. The Metware database (Metware Biotechnology Co., Ltd., Wuhan, China) was utilized to characterize metabolites, and then each substance’s content was quantified based on its peak area. Metabolome data were normalized for principal component analysis [78].

### 4.5. Phytohormone Absolute Quantitation Assay

Seven kinds of phytohormones (a total of 15 substances) were absolutely quantified, including auxin (indole-3-acetic acid), salicylic acid, cytokinin (*trans*-zeatin and *cis*-zeatin), gibberellin (GA-1), ethylene (1-aminocyclopropanecarboxylic acid), abscisic acid (abscisic acid, glucose ester of abscisic acid), and jasmonic acid (12-oxy-plant-dienoic acid, jasmonic acid, methyl jasmonate, jasmonic acid–valine, jasmonic acid–phenylalanine, jasmonic acid–isoleucine, and dihydrojasmonic acid).

Fresh peanut plant root samples frozen at −80 °C were ground into powder and then extracted at 4 °C with 0.5 mL methanol/water/formic acid (15:4:1, *v*/*v*/*v*). The extract was vortexed for 10 min and centrifuged at 4 °C for 5 min at 14,000 rpm. The supernatant was collected, and the above steps were repeated (vortex 5 min and centrifuge 5 min). The combined extracts were evaporated to dryness under nitrogen flow, reconstituted in 80% methanol (*v*/*v*), ultrasonicated (1 min), and filtered (PTFE, 0.22 micron; Anpel), and then LC-MS/MS analysis was conducted. Sample extracts were analyzed using an LC-ESI-MS/MS system (HPLC, Shim-pack UFLC SHIMADZU CBM30A system; https://www.shimadzu.com.cn/ (accessed on 7 August 2022); MS, Applied Biosystems 6500 Triple Quadrupole, http://www.appliedbiosystems.com.cn/ (accessed on 7 August 2022)). The analytical conditions were as follows: HPLC: column, Waters ACQUITY UPLC HSS T3 C18 (1.8 μm, 2.1 mm × 100 mm); solvent system, water (0.04% acetic acid): acetonitrile (0.04% acetic acid); gradient program, 90:10 *v*/*v* at 0 min, 40:60 *v*/*v* at 5.0 min, 40:60 *v*/*v* at 7.0 min, 90:10 *v*/*v* at 7 min, 90:10 *v*/*v* at 10 min; flow rate, 0.35 mL/min; temperature, 40 °C; injection volume: 2 μL. The effluent was alternatively connected to an ESI-triple quadrupole-linear ion trap (Q TRAP)-MS. All of the standards were purchased from Olchemim Ltd. (Olomouc, Czech Republic) and Sigma (St. Louis, MO, USA).

### 4.6. RT-qPCR Assay

Plant root samples used for the RT-qPCR assay were quickly frozen with liquid nitrogen after 48 h treatment and stored in a refrigerator at −80 °C to avoid detection errors caused by RNA degradation. Procedures for the assay can be summarized as follows: RNA was isolated from the samples using a RNAiso Plus kit (TAKARA, Kusatsu, Japan), mRNA was enriched using magnetic beads, and mRNA was purified by hybridization of probes. RNA was reverse transcribed into cDNA using the GoScriptTM Reverse Transcription System (Promega, Beijing, China). An mRNA relative expression analysis was conducted using the QuantStudioTM 5 real-time fluorescent quantitative PCR system (Thermo Fisher Scientific, Waltham, MA, USA). The GoTaq^®^ qPCR Master Mix kit (Promega, Beijing, China) was used to provide cDNA reverse transcriptase and fluorescent dye. Actin-7 (LOC112715878) was selected as the housekeeping gene, and the 2^−ΔΔCt^ method was used to convert the mRNA relative expression. Information on these selected genes (gene ID, homology annotation, primer sequences) is presented in Appendix A.

## 5. Conclusions

Using integrated transcriptome, metabolome, and phytohormone absolute quantification, we investigated the peanut root metabolism regulation by exogenous calcium under salt stress. We found that: (1) exogenous calcium supplementation enhanced the allocation of carbohydrates to the TCA cycle and plant cell wall biosynthesis rather than the shikimate pathway influenced by up-regulating the gene expression of antioxidant enzymes under salt stress; (2) exogenous calcium induced further ABA accumulation under salt stress by up-regulating the gene expression of ABA biosynthesis key enzymes AAO2 and AAO3 while down-regulating ABA glycosylation enzyme UGT71C5 expression; (3) exogenous calcium supplementation under salt stress restored the *trans-*zeatin absolute content to unstressed levels while inhibiting the root *cis*-zeatin biosynthesis.

## Figures and Tables

**Figure 1 plants-12-03130-f001:**
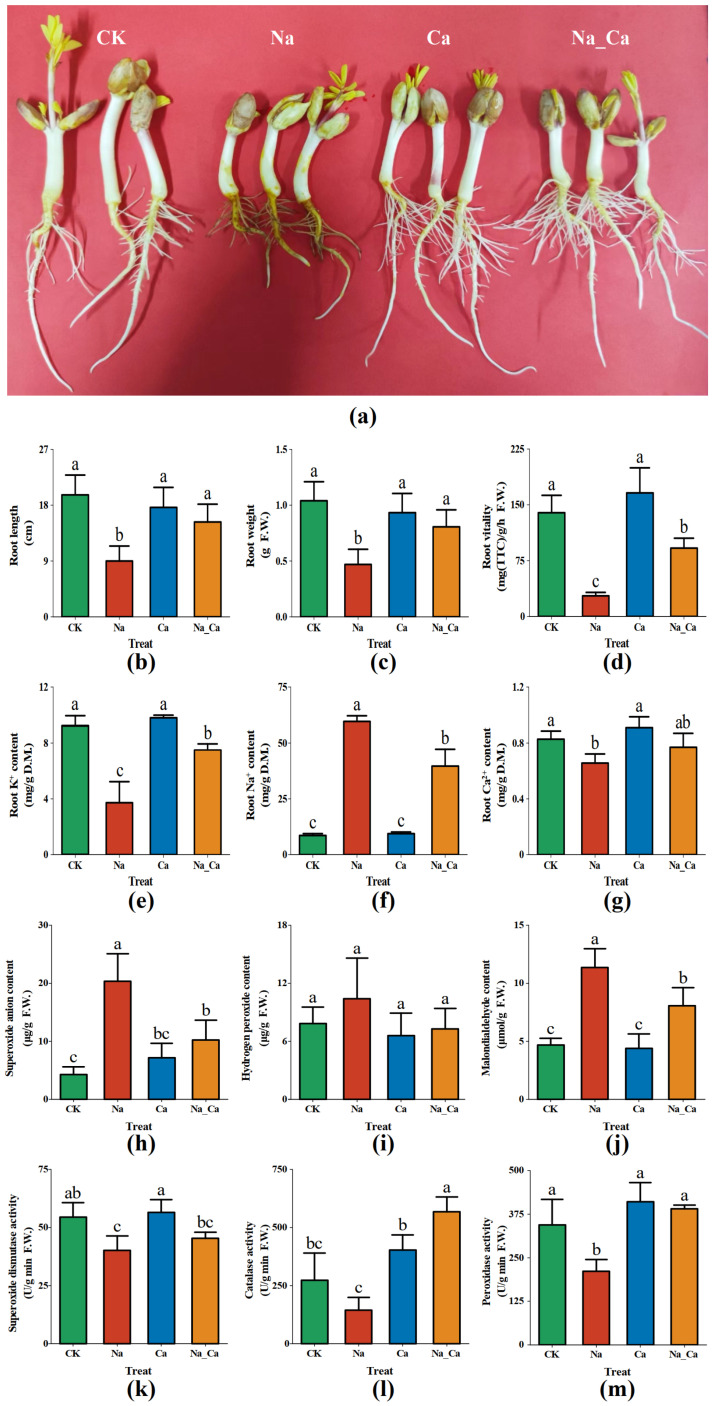
Root growth and physiological phenotypic traits of peanut seedlings after 48 h treatment. (**a**): photo of peanut seedling growth; (**b**): root length; (**c**): root K^+^ content; (**d**): root weight; (**e**): root Na^+^ content; (**f**): root vitality; (**g**): root Ca^2+^ content; (**h**): superoxide anion (O_2_^−^) content; (**i**): hydrogen peroxide (H_2_O_2_) content; (**j**): malondialdehyde (MDA) content; (**k**): superoxide dismutase (SOD) activity; (**l**): catalase (CAT) activity; (**m**): peroxidase (POD) activity. Treatments, CK, untreated; Na, treated with 150 mmol/L NaCl; Ca, treated with 15 mmol/L CaCl_2_; Na_Ca, 150 mmol/L NaCl and 15 mmol/L CaCl_2_ Co-treatment. Each treatment contains 3 biological repeats. The one-way ANOVA and Duncan’s new multiple range method were used to compare the differences between treatments. The significance level is *p* < 0.05. There is no significant difference between the two groups containing the same letters.

**Figure 2 plants-12-03130-f002:**
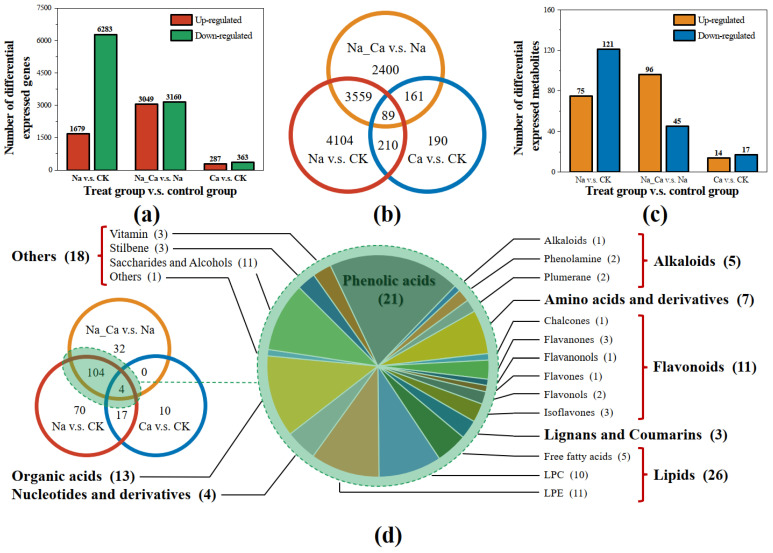
Screening of differentially expressed genes (DEGs) and differentially metabolites (DEMs). (**a**) Differentially expressed genes (DEGs) number in three pairwise comparison groups; (**b**) Venn diagram of differentially expressed genes (DEGs) shared by two pairwise pairwise comparison groups; (**c**) differentially expressed metabolites (DEMs) number in three pairwise comparison groups; (**d**) Venn diagram of differentially expressed metabolites (DEMs) shared by two pairwise comparison groups. Treatments: CK, untreated; Na, treated with 150 mmol/L NaCl; Ca, treated with 15 mmol/L CaCl_2_; Na_Ca, 150 mmol/L NaCl and 15 mmol/L CaCl_2_ Co-treatment. Two pairwise comparison groups: Na vs. CK and Na_Ca vs. Na (experimental group vs. control group).

**Figure 3 plants-12-03130-f003:**
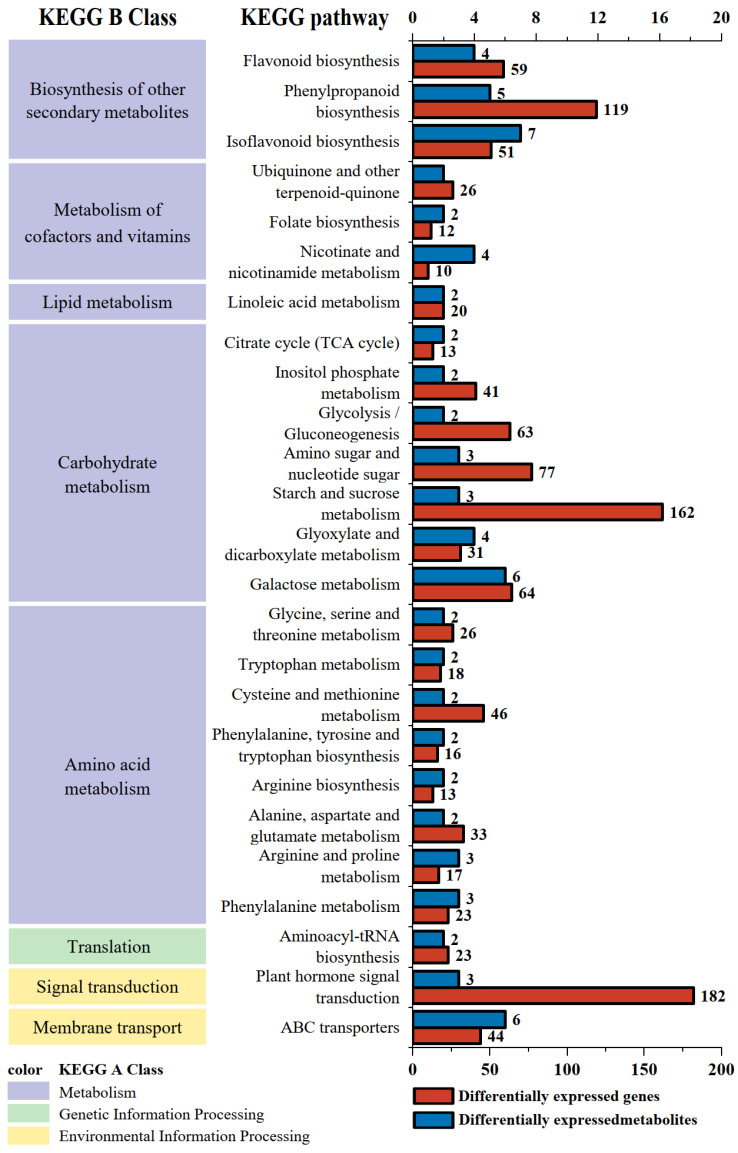
KEGG co-enrichment analysis of differentially expressed genes (DEGs) and differentially expressed metabolites (DEMs) shared by 2 pairwise comparison groups. Treatments: CK, untreated; Na, treated with 150 mmol/L NaCl; Na_Ca, 150 mmol/L NaCl and 15 mmol/L CaCl_2_ co-treatment. Two pairwise comparison groups: Na vs. CK and Na_Ca vs. Na (experimental group vs. control group).

**Figure 4 plants-12-03130-f004:**
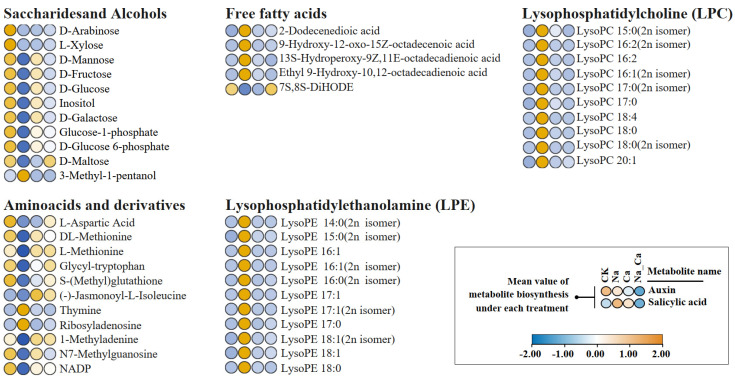
Heatmap of differentially expressed saccharides and alcohols, liquids, amino acids, and derivatives. Treatments: CK, untreated; Na, treated with 150 mmol/L NaCl; Ca, 15 mmol/L CaCl_2_; Na_Ca, 150 mmol/L NaCl and 15 mmol/L CaCl_2_ Co-treatment.

**Figure 5 plants-12-03130-f005:**
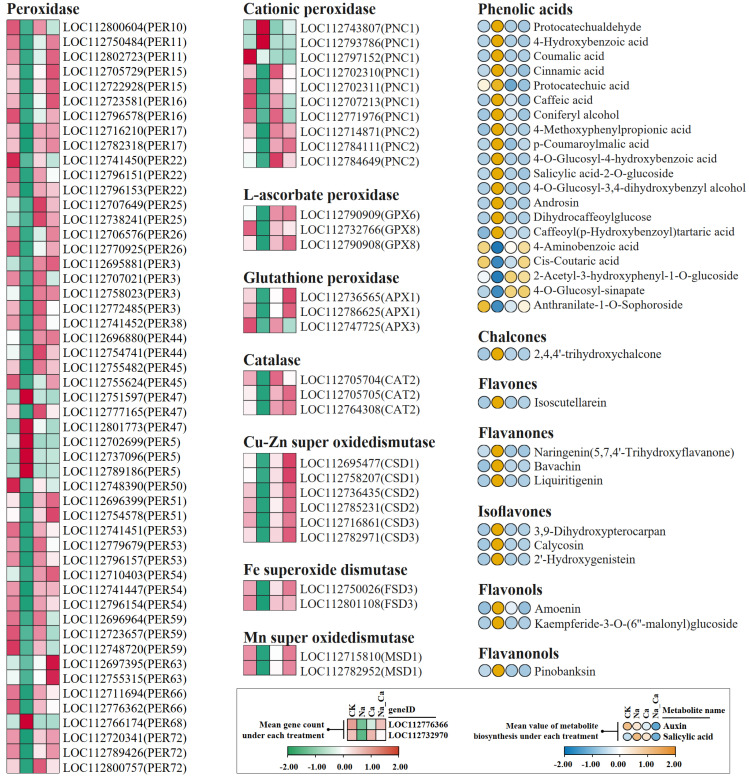
Heatmap of differentially expressed antioxidant enzymes (SOD, POD, CAT) genes expression and differentially expressed C6-C3 secondary metabolites biosynthesis. Treatments: CK, untreated; Na, treated with 150 mmol/L NaCl; Na_Ca, 150 mmol/L NaCl and 15 mmol/L CaCl_2_ Co-treatment.

**Figure 6 plants-12-03130-f006:**
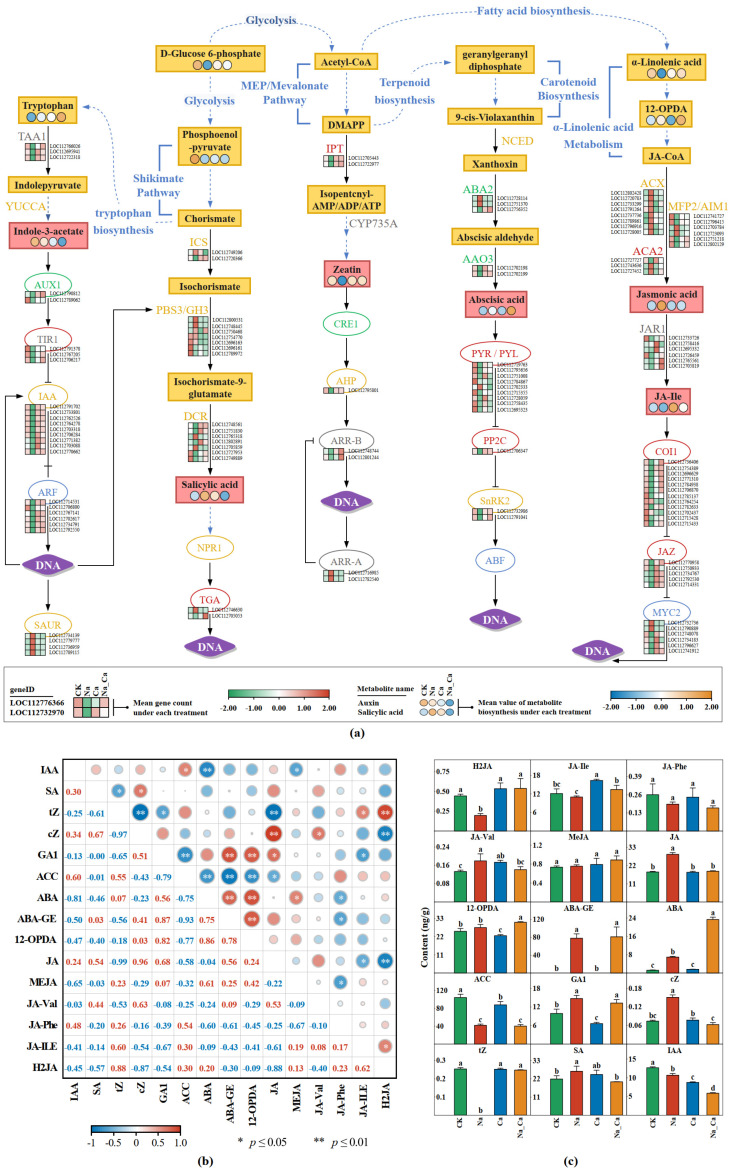
Differentially expressed gene expression and hormone content of hormone biosynthesis and signal transduction under different treatments. (**a**) Heat map of hormone absolute contents and gene expression in hormone biosynthesis and signal transduction pathways; (**b**) Pearson correlation coefficient matrix of 15 plant hormones absolute contents; (**c**) comparison of absolute contents of 15 plant hormones under different treatments. In (**a**) **TAA1**, L-tryptophan-pyruvate aminotransferase 1; **YUCCA**, Indole-3-pyruvate monooxygenase; **AUX1**, auxin transporter protein 1; **TIR1**, protein TRANSPORT INHIBITOR RESPONSE 1; **IAA**, auxin-responsive protein; **ARF**, ADP-ribosylation factor; **SAUR**, auxin-responsive protein; **ICS**, isochorismate synthase; **PBS3**, auxin-responsive GH3 family protein PBS3 (homologous gene, AtGH3.12); **DCR**, BAHD family of acyltransferases (defective in cuticular ridges); **NPR1**, BTB/POZ domain-containing protein NPR1; **TGA**, bZIP DNA-binding family protein transcription factor TGA; **IPT**, adenylate dimethylallyl transferase (cytokinin synthase); **CYP735A**, cytokinin *trans-*hydroxylase CYP35A; CRE1, cytokinin receptor (AHK4); **AHP**, histidine-containing phosphotransferase; **ARR-B**, two-component response regulator ARR-B family protein; **ARR-A**, two-component response regulator ARR-A family protein; **NCED**, 9-*cis*-epoxycarotenoid dioxygenase; **ABA2**, protein abscisic acid deficient 2 (xanthoxin dehydrogenase); **AAO3**, aldehyde oxidase 3 (abscisic–aldehyde oxidase); **PYR/PYL**, abscisic acid receptor PYR or abscisic acid receptor PYL; **PP2C**, protein phosphatase 2C homolog; **SnRK2**, serine/threonine-protein kinase SRK2; **ABF**, abscisic acid responsive elements–binding factor family protein; **ACX**, peroxisomal acyl-coenzyme A oxidase; **MFP2**, peroxisomal fatty acid beta-oxidation multifunctional protein MFP2; **AIM**, peroxisomal fatty acid beta-oxidation multifunctional protein AIM1; **ACA2**, 3-ketoacyl-CoA thiolase 2; **JAR1**, jasmonic acid–amido synthetase JAR1 (auxin-responsive GH3 family protein); **COI1**, c6oronatine-insensitive protein 1; **JAZ**, jasmonate ZIM domain-containing protein; **MYC2**, basic helix–loop–helix (bHLH) DNA-binding family protein transcription factor MYC2. (**b**,**c**) **IAA**, indole-3-acetic acid; **SA**, salicylic acid; ***t*Z**, *trans-*zeatin; ***c*Z**, *cis*-z6eatin; **GA1**, gibberellin A1; **ACC**, 1-a6minocyclopropanecarboxylic acid; **ABA**, abscisic acid; **ABA-GE**, ABA-glucosyl ester; **12-OPDA**, 12-oxophytodienoic acid; **JA**, jasmonic acid; **MeJA**, methyl jasmonate; **JA-Val**, 0; **JA-Phe**, jasmonoyl]-phenalanine; **JA-Ile**, jasmonoyl-L-isoleucine; **H2JA**, dihydrojasmonic acid. Treatments, CK, untreated; Na, treated with 150 mmol/L NaCl; Ca, treated with 15 mmol/L CaCl_2_; Na_Ca, 150 mmol/L NaCl and 15 mmol/L CaCl_2_ Co-treatment. Each treatment contains 3 biological repeats. The one-way ANOVA and Duncan’s new multiple range method were used to compare the differences between treatments. The significance level is *p* < 0.05. There is no significant difference between the two groups containing the same letters.

## Data Availability

The data presented in the study are deposited in Sequence Read Archive (SRA) database at NCBI (SRA BioProject PRJNA964590). Other data of this study can be found in the manuscript or Appendix A.

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
