# Peer review of "Multi-Omics Revealed Peanut Root Metabolism Regulated by Exogenous Calcium under Salt Stress"

_plants, 2023, doi:10.3390/plants12173130_

Round 1

Reviewer 1 Report

The present study reveals how exogenous calcium elevates peanut salt-stress tolerance via three types of omics data including transcriptome, metabolome, and phytohormone. The present data are informative and bring some information in the plant abiotic stress fields. Some main concerns are listed below.

1.      The authors reported that exogenous Ca inhibits the uptake of Na and increases K contents in the roots under salt stress (Figure 1E, F). How exogenous Ca affects the transportation of Na and K in the peanut roots? The authors may dig the related genes expression of Na/K transporters from the omics data.

2.      Salt stress induced ABA accumulation in plants. The present data showed that exogenous Ca induces ABA accumulation in peanut roots further more than under salt stress alone. This maybe not contribute to the exogenous Ca elevating of peanut salt tolerance. 

3.      Multiple omics studies have been applied recently to reveal the mechanisms of plant salt endurance as well as saline-alkaline tolerance in plants. The authors may reference a recent published review paper (Rao et al, 2023 J Plant Physiol, https://doi.org/10.1016/j.jplph.2023.153916) to help readers easily understanding this study.

4. Some errors are needed to be revised. Such as p 0.05 in line 88, Ca+ in line 92. The position of the label of ABCD in each figure panel is weird.  

Author Response

For reviewer 1:
====================

[Reviewer 1, comment 1]
The authors reported that exogenous Ca inhibits the uptake of Na and increases K contents in the roots under salt stress (Figure 1E, F). How exogenous Ca affects the transportation of Na and K in the peanut roots? The authors may dig the related genes expression of Na/K transporters from the omics data.

——————————

[Authors]
Thank you for your comment. 
Exogenous Ca2+ inhibited Na+ uptake and K+ efflux in the roots under salt stress, so we mined the transcriptome data for Na+ and K+ transporters expression and drew a heatmaps by group (Figure S5, corresponding data are in table S7), and discussed them in the discussion section.
----------
Page range in revised manuscript: line 448-464.

====================

[Reviewer 1, comment 2]
Salt stress induced ABA accumulation in plants. The present data showed that exogenous Ca induces ABA accumulation in peanut roots further more than under salt stress alone. This maybe not contribute to the exogenous Ca elevating of peanut salt tolerance.

——————————

[Authors] Thank you for your comment. 

There are many complex aspects involved in plant salt resistance. ABA content alone cannot determine whether it contributes directly to the alleviation of salt stress in plants by exogenous substances.

We also considered:
● The effects of exogenous ABA on salt stress in plants have been reported;
● Some studies have shown that under salt stress, salt-resistant varieties have significantly higher ABA content in roots than salt-sensitive varieties;
● In studies that investigated improving salt resistance in plants by overexpressing calcium repeater genes, overexpressed plants exhibited higher salt resistance and higher levels of ABA under salt stress compared to wild-type plants under salt stress.
● Exogenous calcium was found to increase the ABA content of plants under salt stress alone in some studies. In our study, qPCR data showed that exogenous calcium could up-regulate the key enzyme gene for ABA biosynthesis, AAO2/3, and down-regulate the key enzyme gene for ABA glycosylation, UGT71C5, leading to ABA accumulation.

To summarize, we consider that exogenous calcium-mediated salt stress resistance is comprehensive, and can include, at the very least, the following:
(1) By regulating K+ and Na+ transportation (as mentioned in your first comment);
(2) Up-regulation of antioxidant enzyme genes increases carbohydrates' allocation to the TCA cycle and cell wall biosynthesis rather than secondary metabolism under salt stress (one of the main purposes of our study);
(3) Besides the two points above, the biosynthesis of ABA could contribute to the enhancement of plant salt resistance by exogenous calcium as a signal molecule that can be induced by calcium and mediate plant stress response. (Considering the above references).

----------
Page range in revised manuscript: line 578-597.
Thank you again for your comment.

====================

[Reviewer 1, comment 3]
Multiple omics studies have been applied recently to reveal the mechanisms of plant salt endurance as well as saline-alkaline tolerance in plants. The authors may reference a recent published review paper (Rao et al, 2023 J Plant Physiol, https://doi.org/10.1016/j.jplph.2023.153916) to help readers easily understanding this study.

——————————

[Authors] Thank you for your comment. 

We added a paragraph in the "introduction" section to discuss multi-omics methods application to plant salt resistance studies in light of your opinion, we also added the reference you suggested. It will make our introduction more detailed and provide readers with a better understanding of our work.

----------
Page range in revised manuscript: line 82-95.

====================

[Reviewer 1, comment 4]
Some errors are needed to be revised. Such as p 0.05 in line 88, Ca+ in line 92. The position of the label of ABCD in each figure panel is weird.

——————————

[Authors] Thank you for your comment.
● We apologize for the textual errors in the manuscript. We have made revisions to avoid such mistakes.  
● As for the position of the ABCD label in each figure panel, we adhere to the requirements of the plants magazine Microsoft Word template, which instructs us to place the label below the picture. We corrected the labels of the facets in each figure to be "(lowercase letters)" form.
(https://www.mdpi.com/files/word-templates/plants-template.dot).

====================

Thank you again for all your comments on this manuscript!
Yours sincerely,
Corresponding author: Guolin Lin
E-mail: [email protected]

Reviewer 2 Report

In the Ms Multi-omics revealed peanut root metabolism regulated by exogenous calcium under salt stress, authors integrated 3 types of omics data (transcriptome, metabolome, and phytohormone absolute quantification) to analyze the metabolic profiles of peanut seedling roots as regulated by exogenous calcium under salt stress. The study is interesting, however, the Ms should be thoroughly revised before acceptance.

Abstract, Line 17, 20, 22, Exogenous or exogenous ?

Graphical abstract, Most of the things are not visible.

Introduction is very haphazardly  written. It should be focused at the study. The first para should be related to Ca signalling and impact. Author may see a recent book ‘Calcium transport elements in plants’ for details. What are the components (given in chapter one), how it affect the plants under stress and its application (last chapter).  

Then, they should discuss the impact of salt stress in the next para, thereafter, they can discuss the application of Ca in salt stress management. There are a lot of studies available at the subject, authors should review most of the related content. Further a few recent studies elaborated indigenous methods for Ca increase in plants by overexpressing a Ca transporter, TaNCL2-A in Arabidopsis, that also showed the salt tolerance. These kinds of studies should be included in introduction. Then author may reach to the gap area, why this particular study has been performed.

The study has a lot of data but the results are again not arranged smartly. There is no need of confusing figures. Some of them can be shifted to the suppl fig. Figure legends should be properly described. Results in some sections are also seems to be descriptive only, that needs to be revised thoroughly.

Discuss seems to be full of results. It should be focussed on only important findings in the results, not the repetition of results.

Change in various antioxidant enzymes like SOD, GPX, APX, Catalase, GRs has been reported in numerous studies under salt stress. Several recent studies in bread wheat have shown the changes in these enzymes and in the expression of their encoding genes in the presence of salt stress (for ex. https://link.springer.com/article/10.1007/s00299-021-02717-1). The authors have shown the results, but nothing is in discussion. They may discuss them properly in comparison to the other plants like bread wheat and rice antioxidant genes.

Minor editing is required. 

Author Response

For reviewer 2:
====================

[Reviewer 2, comment 1]
Abstract, Line 17, 20, 22, Exogenous or exogenous?

——————————

[Authors] Thank you for your comment. 
We apologize for the textual errors in the manuscript. We have made revisions to avoid such mistakes. We apologize again for our mistake.

====================

[Reviewer 2, comment 2]
Graphical abstract, most of the things are not visible.

——————————

[Authors] Thank you for your comment. 
Our apologies for any inconvenience caused by the Graphical abstract being copied to Word with a resolution too low to be clearly viewed. This time, “Graphical Abstract.pdf” was incorporated into “revised_manuscript_20230824.zip”. 
Please accept our sincere apologies for it.

====================

[Reviewer 2, comment 3]
Introduction is very haphazardly written. It should be focused at the study. The first para should be related to Ca signaling and impact. Author may see a recent book ‘Calcium transport elements in plants’ for details. What are the components (given in chapter one), how it affects the plants under stress and its application (last chapter). 

——————————

[Authors] Thank you for your comment. 
Thank you for your comment.

The suggestion is constructive. 

Meanwhile, we considered:
●    For this study, we aim to investigate the regulation of exogenous calcium on plant root metabolism under salt stress. So, in this revision, we also stated about "salt stress affects plant metabolism and hormone biosynthesis" in the first paragraph "Effects of salt stress on plants", which should be directly related to our topic.
●    At the same time, we have also read some studies on "exogenous calcium relieves salt stress in plants ". In most of them, the first paragraph of the introduction introduces "the hazards of salt stress", then the second paragraph describes "the role of exogenous calcium on salt stress", then the knowledge gap is pointed out to introduce the study.
●    Of course, there are also many "studies on abiotic stress mitigation by exogenous substances" that include the significance of exogenous substances in the first paragraph, and then introduce the abiotic stress of the study in the second paragraph. We have also tried to start the introduction in this way and have written an introduction version based on your comment. However, we found that it is difficult for us to use a paragraph to summarize such a huge knowledge structure like "calcium signal" or "calcium transport element". And the first paragraph will not be able to directly point "salt stress affects plant metabolism" if this is the case.

So, the first two paragraphs of the introduction were revised as follows: 

(1)paragraph 1 describes the effects of salt stress followed by a reference to the effects of salt stress on metabolism and phytohormones.
(2)paragraph 2 we tried to introduce at 4 levels:
① calcium fertilizer with chemical fertilizer promotes crop growth and increases yield;
② exogenous calcium improves plant antioxidant enzyme activity, reduces Na+ uptake and K+ loss under salt stress.
③ overexpression of calcium signal repeater protein coding gene to improve plant salt resistance (inspired by your comment);
④ overexpression of calcium transporter genes improves plant salt stress resistance. In addition to citing the literature you suggested to us we have also cited some other related studies (based on your comment).

These studies directly or indirectly proved the importance of calcium in improving plant salt stress resistance.

----------

The book you recommended broadened our horizons. Paragraph 2 contains a descriptive sentence: "Some studies also reported that overexpression of genes encoding for calcium signaling repeater proteins or genes encoding for calcium transporter elements can improve salt resistance in plants." cited the literature you recommended (ref. 17). We will also refer to this valuable book in our subsequent research.

----------
There is also an introduction version (in which the first paragraph introduces "calcium transport elements") through either of the two links below. If you prefer this version (introduction_20230822), we'd like you to point it out in your comments, and we'll try to revise in the next version of introduction.

https://1drv.ms/w/s!AoHo_isqVl69hxjHEO-bI-uwZElA?e=E9Off1
https://r2phr2inkl.feishu.cn/docx/XkSedN6RSoQieix7IbOczbGbnVg?from=from_copylink

----------
Thanks again for your suggestion.

Line range in revised manuscript: line 37-70.

====================

[Reviewer 2, comment 4]
Then, they should discuss the impact of salt stress in the next para, thereafter, they can discuss the application of Ca in salt stress management. There are a lot of studies available at the subject, authors should review most of the related content. Further a few recent studies elaborated indigenous methods for Ca increase in plants by overexpressing a Ca transporter, TaNCL2-A in Arabidopsis, that also showed the salt tolerance. These kinds of studies should be included in introduction. Then author may reach to the gap area, why this particular study has been performed.

——————————

[Authors] Thank you for your comment. 
We have revised the relevant part of the introduction in light of your opinion. We cited the reference you suggested and other recent studies on overexpressing Ca2+ transporters to improve plant salt stress resistance in the second paragraph.

----------
Page range in revised manuscript: line 65-67.

====================

[Reviewer 2, comment 5]
(1) The study has a lot of data but the results are again not arranged smartly. There is no need of confusing figures. Some of them can be shifted to the suppl fig. Figure legends should be properly described.
(2) Results in some sections are also seems to be descriptive only, that needs to be revised thoroughly.

——————————

[Authors] Thank you for your comment. 

(1) The revised manuscript consists of 6 figures (8 figures in the pre-revised manuscript), and the “transcriptome GO enrichment analysis” has been set to "Figure S3", Figure 1 and Figure 2 in the un-revised manuscript, were set to “Figure 1” after the revision. Meanwhile, we found some errors when checking the figure captions and caption text. We tried to correct them to make them look more descriptive of the figure content. Thank you again for your careful guidance. We will be more careful in our future academic careers and try to avoid similar errors.

----------
Page range in revised manuscript: 
● line 153-162;
● line 213-220;
● line 247-251;
● line 279-282;
● line 306-309;
● line 386-418.

——————————

(2) Results in the unrevised version of the manuscript are only descriptive and we have attempted to briefly summarize "what the above results illustrate" after every result. Thanks for the suggestion, it makes the article results more structured and readable.

----------
Page range in revised manuscript: 
● line 148-151;
● line 208-211;
● line 241-246;
● line 274-278;
● line 300-304;
● line 335-338;
● line 362-366;
● line 381-383;
● line 439-445.

====================

[Reviewer 2, comment 6]
Discuss seems to be full of results. It should be focused on only important findings in the results, not the repetition of results.

——————————

[Authors] Thank you for your comment. 
In the discussion section of the manuscript, we have removed a lot of repetitions of results and tried to re-discuss some important results comprehensively and provide our viewpoint on them.

----------
Page range in revised manuscript: 
●line 468-471;
●line 512-517;
●line 525-528;
●line 539-542;
●line 558-561;
●line 574-578.

====================

[Reviewer 2, comment 7]
Change in various antioxidant enzymes like SOD, GPX, APX, Catalase, GRs has been reported in numerous studies under salt stress. Several recent studies in bread wheat have shown the changes in these enzymes and in the expression of their encoding genes in the presence of salt stress (for ex. https://link.springer.com/article/10.1007/s00299-021-02717-1). The authors have shown the results, but nothing is in discussion. They may discuss them properly in comparison to the other plants like bread wheat and rice antioxidant genes.

——————————

[Authors] Thank you for your comments. 
This part of discussion has been revised. Besides the article you suggested to us, we also cited other relevant references in this paragraph to discuss that overexpression of antioxidant enzyme genes can improve plant salt tolerance. Moreover, we attempted to discuss the possibility of exogenous calcium up-regulation of antioxidant enzyme genes under salt stress by citing literature and transcriptome data mining.

----------
Page range in revised manuscript: line 465-483.

====================

Thank you again for all your comments on this manuscript!
Yours sincerely,
Corresponding author: Guolin Lin
E-mail: [email protected]

Round 2

Reviewer 2 Report

Language editing is required. 

Language editing is required.